# Automated Interlaboratory Comparison of Therapeutic Drug Monitoring Data and Its Use for Evaluation of Published Therapeutic Reference Ranges

**DOI:** 10.3390/pharmaceutics15020673

**Published:** 2023-02-16

**Authors:** Jens Borggaard Larsen, Elke Hoffmann-Lücke, Per Hersom Aaslo, Niklas Rye Jørgensen, Eva Greibe

**Affiliations:** 1The Datasupport Centre for Personalised Medicine, Region South, DK 5000 Odense, Denmark; 2Department of Clinical Biochemistry, Aarhus University Hospital, DK 8200 Aarhus, Denmark; 3Institute for Clinical Medicine, Health, Aarhus University, DK 8000 Aarhus, Denmark; 4The Laboratory of the Danish Epilepsy Centre–Filadelfia, DK 4293 Dianalund, Denmark; 5Department of Clinical Biochemistry, Rigshospitalet, DK 2600 Glostrup, Denmark; 6Department of Clinical Medicine, Faculty of Health and Medical Sciences, University of Copenhagen, DK 1165 Copenhagen, Denmark

**Keywords:** therapeutic drug monitoring (TDM), therapeutic reference range, antidepressant, antipsychotic, big data

## Abstract

Therapeutic drug monitoring is a tool for optimising the pharmacological treatment of diseases where the therapeutic effect is difficult to measure or monitor. Therapeutic reference ranges and dose-effect relation are the main requirements for this drug titration tool. Defining and updating therapeutic reference ranges are difficult, and there is no standardised method for the calculation and clinical qualification of these. The study presents a basic model for validating and selecting routine laboratory data. The programmed algorithm was applied on data sets of antidepressants and antipsychotics from three public hospitals in Denmark. Therapeutic analytical ranges were compared with the published therapeutic reference ranges by the Arbeitsgemeinschaft für Neuropsychopharmakologie und Pharmakopsychiatrie (AGNP) and in additional literature. For most of the drugs, the calculated therapeutic analytical ranges showed good concordance between the laboratories and to published therapeutic reference ranges. The exceptions were flupentixol, haloperidol, paroxetine, perphenazine, and venlafaxine + o-desmethyl-venlafaxine (total plasma concentration), where the range was considerably higher for the laboratory data, while the calculated range of desipramine, sertraline, ziprasidone, and zuclopenthixol was considerably lower. In most cases, we identified additional literature supporting our data, highlighting the need of a critical re-examination of current therapeutic reference ranges in Denmark. An automated approach can aid in the evaluation of current and future therapeutic reference ranges by providing additional information based on big data from multiple laboratories.

## 1. Introduction

Therapeutic drug monitoring (TDM) is a clinical tool where the concentration of a pharmaceutical drug in a biological matrix is used to optimise and individualise the treatment of a patient. TDM is applicable for drugs having a narrow therapeutic range; a correlation between the blood concentration and the therapeutic effect or adverse reactions; and where the symptoms of the disease are difficult to monitor or quantify [1,2,3]. This is particularly true for the treatment of many neurological and psychiatric disorders, but TDM is also used for monitoring immunosuppressive drugs and antibiotics [4,5].

An important parameter for the successful clinical application of TDM is the availability of an accurate therapeutic reference range. Therapeutic drug monitoring can be applied for optimising the dosage and hereby treatment with a drug, using two strategies [2,4,5,6]. If a well-defined therapeutic reference range is available, dose titration can be performed until the patient is within this range. During treatment, any relapse of the disease or acquired adverse reactions should trigger a new measurement and adjustment of dosage accordingly, within the range [3].

As an alternative to the use of a common therapeutic reference range, a patient can be used as his or her own reference [6]. In this individual approach, the dosing of the patient is titrated until the clinician judges the treatment to be optimal, with a minimum of adverse effects. A measurement of the plasma concentration taken at this point in time will serve as the patient’s future reference and used as a target for dose adjustments should changes in the treatment arise. Although, a therapeutic reference range using this method is not strictly necessary, it still holds considerable value as a guidance of the therapy and allows the optimal dosing of the patient to be related to a toxic alarm limit [6].

The therapeutic reference range is defined by a lower concentration of the drug where the treatment is starting to take effect and an upper where the increase in the therapeutic effect declines, and there is an increased chance of the patients’ acquiring adverse drug reactions. The reference range should therefore mirror the dosage span, containing the optimal therapeutic effect for most of the patients. As in the case of endogen substances, the plasma concentration of a drug varies considerably, even when adjusting for dosage. This is primarily due to environmental, physiological, and genetic factors. However, compared with an endogen substance, there are important differences when defining the reference range for a therapeutic drug [7]:

The plasma level of a TDM drug is dosage dependent. The lowest measurement is zero (or the limit of quantification of the laboratory analysis), while the upper depends on the prescribed dosage.Defining a reference range for an endogenous substance is carried out to identify the variance in the healthy population. In contrast, an optimal therapeutic reference range should be based on the variation in a sick but, in regard to the therapeutic effect of the drug, well-treated cohort.The same drug may be used for the treatment of different diseases, having different biochemical origins, symptoms, and therapeutic ranges. This is particularly true for psychopharmacologic treatments.

The calculation of a therapeutic reference range can further be made challenging because of a limited population size [7]. This is especially true when it comes to the treatment of rare diseases or to not commonly prescribed drugs. Because of this, the published therapeutic reference ranges of psychoactive drugs are often based on a limited number of specifically selected patients, which does not necessarily reflect the variations in the real population [6]. Small population sizes and large individual variation may also explain a lack of correlation between the blood concentration and therapeutic effect found in some studies [3].

An alternative to a controlled population, when defining the therapeutic reference range of a drug, is the use of retrospective data from laboratory information management systems (LIMS) [8,9,10]. The benefit of this is a larger data set and the capability of being able to relate drug plasma concentration with adverse effects, e.g., neutropenia or liver function. However, normally, it comes with no information on dosage, the clinical reasoning for ordering the measurement, or the therapeutic effect of the treatment. Laboratory data sets also commonly hold numerous results from patients that are excluded in a controlled setup.

The Arbeitsgemeinschaft für Neuropsychopharmakologie und Pharmakopsychiatrie (AGNP) published the “Consensus Guidelines for Therapeutic Drug Monitoring in Neuropsychopharmacology” [3], which has become the primary reference within the field of TDM. This work evaluates indications for the use of drug monitoring for psychiatric drugs and presents therapeutic reference ranges that are based on the current literature. The guideline is an impressive and important contribution that lists more than 1300 references, and it is currently in its third edition. However, one point of concern regarding this work is the validity of some of the older citations used to support the ranges herein and the transparency of the calculation of those that have been concatenated from several studies [11]. Although the variation in the precision between the laboratories has significantly improved over the past 20 years, most published ranges date back to a few early studies performed when the drug was first introduced to the market. Commonly, these studies relied on a single laboratory test developed in-house, and verification using modern techniques is therefore necessary. The continual re-evaluation of therapeutic reference ranges is also important as the treatment paradigm of a drug may change over time [12]. For these reasons, a ratification is warranted before a published range can be used in the clinic, and the agreement with current laboratory methods needs to be ensured.

In this study, we present a model for sorting and selecting routine TDM data from laboratory LIMS systems. The algorithm of the model has been integrated into the software RefIT, which facilitates the fast and easy batch calculation of analytical ranges that are based on percentiles of the sorted data sets. The software was applied on data sets from a TDM analysis of antidepressants and antipsychotics, from three public hospitals in Denmark. The results allow a comparison between the actual laboratory findings and the therapeutic reference ranges published in the AGNP consensus guideline [3].

## 2. Materials and Methods

### 2.1. Model Description

To avoid confusion between the therapeutic reference ranges, which have been clinically validated, and the ones calculated on the basis of retrospective laboratory data in this study, throughout the manuscript, we use the term “therapeutic analytical range” for the latter. This, because we find it a more correct term than “concentration in blood”, used by Hiemke et al. [3].

Laboratory LIMS data may hold several subpopulations that are otherwise omitted when calculating therapeutic reference ranges on the basis of controlled experimental setups. These subpopulations are not outliers in a laboratory sense. Rather, they are samples or patients that would normally be excluded before performing the calculation. The individual subpopulations may consist of the following:Patients that have not been medication fasting prior to sampling and therefore are not at their minimum at steady state.Patients that have not reached steady state at the time of the sampling.Samples taken during adjustment of dosage and where the patient is not optimally treated.Samples from patients not taking the medicine as prescribed (noncompliance).Patients that are abusing the medicine.Samples from patients with a high degree of comedication or who are taking the medicine in combination with general drug abuse.Patients receiving a standard dosage but who are pharmacogenetically poor or ultrafast metabolisers of the drug.Patients that are misdiagnosed and thus cannot be optimally treated.

As there is no information on the health status or therapeutic effect of a patient at the sampling time in the LIMS system, there is no direct way of validating results for calculating a therapeutic analytical range. One option is to include all samples. However, this would include all the subpopulations in the data set. In contrast, selection between the results from a patient can be performed by either taking the first or last of the measurement. These methods all weight each of the patients equally: both patients that never obtain optimal treatment and stable patients in long-term treatment.

The presented model for evaluating and excluding such subpopulations from retrospective laboratory data sets relies on the premise that the TDM measurements are requested by trained clinicians, following common practice for ordering this type of analysis (Figure 1). Although it provides a very simplified version of the clinical decision-making, it facilitates the development of an algorithm capable of objectively sorting and selecting large numbers of data. A flowchart of the model is shown in Figure 1. With no information in the LIMS data on why a sample has been requested, the selective parameter of the model is set as the time interval between sequential measurements. The main assumption is that TDM is requested to check the patient for compliance or for optimising the therapeutic treatment using the common or individual range approach, described in the introduction of this paper. For all these three scenarios, it is anticipated that any discrepancy in the result will be followed by clinical intervention and an additional measurement from the patient. A single measurement will in this regard either suggest a check for compliance, a sample taken as reference, or fit within a published therapeutic reference range. If there is no immediate additional sequential sample from the same patient, this would suggest that the result was approved and thus can be included in the data set for calculating the therapeutic analytical range. However, if the result is followed shortly after by a second measurement, this would indicate that the first is obsolete and should be excluded from the analysis (Figure 1).

Alternatively, if the time between two samples is long (months), this suggests that the patient was optimally treated during the interval. The second measurement may be requested either on the background of a routine consultancy or because of a change in the treatment. In the first case, this second result is included to ensure that optimally treated patients are weighted higher in the data set when calculating the therapeutic analytical range. The latter would be considered as a new treatment and be included depending on the presence of further sequential samples (Figure 1).

### 2.2. Data Collection

Retrospective results from TDM analysis of antidepressants and antipsychotics were extracted from the LIMS system of the three participating Danish laboratories. These were the Department of Clinical Biochemistry, Aarhus University Hospital, Aarhus (AUH); the Laboratory of the Danish Epilepsy Centre–Filadelfia, Dianalund (EHL); and the Department of Clinical Biochemistry, Rigshospitalet (RH), Copenhagen. Approvals were obtained from the respective hospital boards, prior to downloading the data.

Therapeutic drug monitoring at AUH is carried out by using LC-MS/MS technology and assays that have been developed in-house. The analyses are all accredited according to ISO:15189:2013, and the quality is externally monitored by proficiency testing. Results were extracted for the following drugs covering a period from 1 January 2014 to 31 December 2018: amitriptyline/nortriptyline (metabolite), aripiprazole/dehydroaripiprazole (metabolite), citalopram/escitalopram, clomipramine/desmethylclomipramine (metabolite), clozapine, duloxetine, fluoxetine/norfluoxetine (metabolite), imipramine/desipramine (metabolite), mirtazapine, olanzapine, perphenazine, quetiapine, risperidone/paliperidone (metabolite), sertraline, venlafaxine/o-desmethyl-venlafaxine (metabolite), ziprasidone, and zuclopenthixol.

All the assays performed at EHL have been developed in-house by using LC-MS/MS technology. These include the same drugs as are analysed at AUH and in addition flupentixol, haloperidol, and paroxetine. Although part of the laboratory production, fluoxetine/norfluoxetine, mianserine, and ziprasidone were excluded, owing to a low number of samples (<100). The assays (*n* = 26) are accredited according to ISO15189:2013, with the exception of aripiprazole/dehydroaripiprazole, and mirtazapine. The quality is externally monitored by proficiency-testing programmes, covering all analytes. For this study, data were collected from the laboratory LIMS system spanning a period from 1 January 2012 to 30 March 2022.

The analyses for therapeutic drugs at RH is performed by HPLC using UV-detection. The following drugs are included in the laboratory repertoire: amitriptyline/nortriptyline, clomipramine/desmethylclomipramine, clozapine, dosulipine/northiaden, and imipramine/desipramine, of which amitriptyline, nortriptyline, clozapine, and imipramine/desipramine are accredited according to ISO 15189:2013. The quality of the assays is monitored by external proficiency-testing schemes. For the calculations, data were collected covering the period from 9 May 2011 to 26 April 2022.

### 2.3. Data Analysis and Evaluation of the Model

The model for sorting and selecting the retrospective data sets has been incorporated into the software RefIT (available from GitHub under a general public license (GPL)—download address: https://github.com/JensLarsen/RefIT/releases/tag/V1.0). For evaluation purposes, RefIT has additional functions for calculating a therapeutic analytical range using one data point per patient (either the first or the last entry) and by including all samples in the data set. Percentiles are set in the software, which also has the selection of the different mathematical methods for calculating them. Other built-in features are the selection of sex and age interval, the removal of outliers on the basis of Tukey’s fences, and setting the minimum time for including two sequential samples in the TDM model. The software takes data input from Excel files and provides data export for the results to the same file format for easy validation. In addition, percentile and normal distribution graphs can be exported as image files.

The TDM model for selecting data and calculating therapeutic analytical ranges was evaluated by comparing the four models of data selection in the software. This was carried out on a single data set from AUH, using data from the drugs clozapine, perphenazine, and imipramine, covering high (>2000), mid (>500), and low (<200) numbers of data points in the data set.

### 2.4. Calculation of Therapeutic Analytical Ranges

The RefIT software was used on data sets obtained from the LIMS systems of the participating laboratories. These are situated in three of the five regions of Denmark, and uses two different LIMS software. At AUH and RH, Labka II is used (CSC Denmark A/S, Copenhagen, Denmark), while EHL uses LIMS software BCC (CGI Inc., Ballerup, Denmark). Both LIMSs facilitate the export of data as Excel files that can be imported directly into RefIT.

Data were selected by setting the period for routine consultancy in the model to 7 months. All calculations were performed while not removing outliers (Tukey’s fences deselected in the RefIT software). To increase the robustness of the ranges, a limit of >100 samples was set for calculating them. Therapeutic analytical ranges were calculated using the selected data from each of the three laboratories, as 10–90 and 25–75 percentiles. General statistics, including the total number of samples and samples included in the analysis, are provided as Appendix A to this paper.

For each analyte, a combined and a sex-specific therapeutic analytical range was calculated. To investigate the effect of age, the intervals 20–64 years and 65–100 years were selected. Age- and sex-specific differences were calculated on the basis of the data set supplied by AUH.

The lower and upper percentile limits from each compound were compared between the individual laboratories. This was done by allowing a maximum bias of 30% (+/−15%) compared with the average value between the laboratories.

## 3. Results

In total, 199,964 measurements of 31 antidepressant and antipsychotic drugs were collected from the LIMSs of the three participating laboratories. Applying the TDM model for the selection of data led to the inclusion of 111,300 data points from 79,119 patients (see Appendix A). As the repertoire differs between the laboratories, not all the analytes are represented in each data set.

### 3.1. Evaluation of the TDM Model

The TDM model for selecting data (Figure 1) was evaluated by comparing the calculated therapeutic analytical range with the ranges obtained by including all samples in the data set and by using one point per patient, defined as either the first or the last sample in the series. Figure 2 shows the complete data set of clozapine from AUH, and Figure 3 shows the four calculated percentile plots. Table 1 shows the therapeutic analytical ranges of clozapine, perphenazine, and imipramine obtained using all of the four methods. Including all the data points resulted in a much wider therapeutic analytical range for clozapine and imipramine, while using only the first data point from each patient resulted in a lower calculated range. Including only the last entry from each patient resulted in a calculated range closer to that of the TDM model, but this was based on a lower number of samples.

### 3.2. Interlaboratory Comparison

In general, there was good concordance between the calculated therapeutic analytical ranges of the participating laboratories (Table 2). The lower 10% percentile limit deviated more than was the case of the upper 90% level. For three of the analytes, there was disagreement (>30% bias) in regard to the upper 90% limit. These were citalopram/escitalopram (same LC-MS/MS analysis as escitalopram is the s-enantiomer of citalopram), quetiapine, and mirtazapine. Here, the data sets from EHL showed a higher calculated therapeutic analytical range than that of AUH.

### 3.3. Comparison to Published Therapeutic Reference Ranges and Ranges in the AGNP Consensus Guideline

The calculated therapeutic analytical ranges were compared with the reference ranges listed in the “Consensus Guidelines for Therapeutic Drug Monitoring in Neuropsychopharmacology” and the citations given here in [3,13,14,15,16,17,18]. As many of the ranges in the guideline are concatenated from multiple studies, with no additional information on the primary references and the calculation method, both 10–90 (Table 2) and 25–75 (Table 3) percentiles were calculated for each drug. For most of the ranges, there were a better correlation when the analytical range was calculated as a 10–90 percentile (Table 2). Although some variations existed between the laboratories, the 10% lower limit of the calculated therapeutic analytical ranges was, for most analytes, below those reported in the AGNP guideline. This was true for all except the drugs flupentixol, haloperidol, perphenazine, sertraline, and venlafaxine + o-desmethyl-venlafaxine (total) (Table 2). The upper 90% limit calculated for these compounds was also significantly higher than the published therapeutic reference ranges of the AGNP.

For perphenazine, the reported range of the AGNP consensus guideline is 1.5–6.0 nmol/L, with the main reference being a review by Putten et al. [19]. This review cites two studies from the early 1980s, defining a lower limit of therapeutic effect at 1.5 nmol/L, with adverse effects observed at a plasma concentration of >3 nmol/L. The therapeutic analytical ranges calculated from multiple laboratories in our study indicate a higher range, similar to that published by Kistrup et al. at 1.8–18 nmol/L (Table 2) [20].

The published therapeutic reference range for the antipsychotic drug haloperidol is from 2.7 to 26.6 nmol/L [3]. However, several studies not cited in the AGNP guideline report a larger range and a higher upper limit: Morselli et al. 2.66–39.9 nmol/L [21]; 5.3–63.8 nmol/L [22]; 26.6–66.5 nmol/L [23,24]; and Palao et al. 14.6–38 nmol/L [25]; 14.9–44.9 nmol/L [26]; 13.3–45 nmol/L [27]. In addition, Putten et al. cites four other studies [19]. The therapeutic analytical range from 5 to 48 nmol/L that was calculated in our study supports these other findings (Table 2).

The AGNP cites four references for the presented therapeutic reference range for flupentixol, denoting it to be from 1.2 to 11.5 nmol/L [3]. The main citation is a study from 1985 by Balant-Gorgia et al., who calculated a threshold for antipsychotic response to 4.7 nmol/L [28]. A more recent study of those cited was by Roman et al. 2008, who presented a range from 2.3 to 34.5 nmol/L [29]. This was similar to the one defined by Kistrup et al. from 1.2 to 37 nmol/L [20]. Together with the therapeutic analytical range from 2 to 23 nmol/L calculated here (Table 2), these studies indicate that the upper limit of flupentixol should be higher than that reported by AGNP.

The published therapeutic reference range for paroxetine is reported to be from 61 to 198 nmol/L [3]. This range seems to be concatenated mainly from two studies: one by Gex Fabry et al., indicating an optimal plasma level from 21 to 198 nmol/L, and one by Tomita et al., reporting a range from 61 to 182 nmol/L [30,31]. The therapeutic analytical range calculated in the current study is considerably higher, ranging from 61 to 537 nmol/L (Table 2). However, this result stems from a single laboratory and has been calculated on the basis of a relatively low number of samples (*n* = 324). However, Reis et al., in a similar population-based study, reported a comparably high range for paroxetine: 29–433 nmol/L [8].

The AGNP guideline cites 12 studies for the therapeutic reference range for venlafaxine and its active metabolite *o*-desmethyl-venlafaxine [3]. The total range for the combined substances is given as 361–1520 nmol/L. Although the exact origin for this range is not indicated, this is very close to the upper limit reported by Shams et al., 685–1534 nmol/L, and Veefkind et al., 722–1482 nmol/L, which are two of the cited papers [32,33]. However, Shams et al. used 25–75 percentiles, which may be the primary reason why the upper limit is considerably lower than the therapeutic analytical ranges in our study, which is on average from 550 to 2555 nmol/L when applying 10–90 percentiles (Table 2). A higher upper limit than the one presented in the guideline was suggested by Reis et al. [8], and Scherf-Clavel et al. also argued for a higher therapeutic reference range for the total moiety of venlafaxine and the active metabolite o-desmethyl-venlafaxine [12].

Opposite to the drugs mentioned above, four others were identified who had calculated therapeutic analytical ranges that were considerably lower than those reported in the AGNP consensus guideline. These were that of the antidepressants desipramine and sertraline and the antipsychotics ziprasidone and zuclopenthixol (Table 2).

The AGNP cites a single reference as the source of the range for desipramine, defined as being from 375 to 1125 nmol/L [3]. However, this study by Pedersen et al. focused on self-intoxicated patients, thus defining the upper limit of the range as the same as the toxic alarm level [3,34]. Desipramine has nonlinear kinetics, and although two studies by Nelson established a level of response at 431 nmol/L, to our knowledge, there is no study defining a complete therapeutic reference range [35,36,37]. According to the calculated therapeutic analytical range presented here, from 47 to 460 nmol/L (average from all participating laboratories), most patients would be below the responsive level if treated with desipramine alone (Table 2).

The range presented by the AGNP for sertraline is from 33 to 491 nmol/L [3]. In contrast, the two laboratory data sets included in our study both gave a calculated therapeutic analytical range <300 nmol/L (average 35–240 nmol/L; see Table 2). In a comparative study by Reis et al., also based on naturalistic data, a therapeutic analytical range was calculated as 19–182 nmol/L [8].

The therapeutic reference range defined in the AGNP consensus guideline for ziprasidone is from 128 to 510 nmol/L [3]. This seems to be based primarily on the two references by Cherma et al. and Vogel et al., who defined a range from 209 to 479 nmol/L and from 128 to 332 nmol/L, respectively [9,38]. Both of these reported results were based on naturalistic data and calculated as 25–75 percentiles. The calculated therapeutic analytical range in this study is considerably smaller, from 77–237 nmol/L, when performed using similar 25–75% fractions (Table 3—38–347 nmol/L with 10–90 percentile range). It should in this regard be noted that data only were available from one laboratory and at a relatively low number (*n* = 375). Regenthal et al., however, listed an even-lower therapeutic range for ziprasidone: from 51 to 153 nmol/L [39].

The therapeutic range reported by the AGNP for zuclopenthixol is from 10 to 125 nmol/L [3]. In contrast, the calculated therapeutic analytical range in our study, obtained from two separate laboratory data sets, gave a maximum range from 6 to 62 nmol/L (Table 2). In a comparable study by Jonsson et al., also based on routine laboratory data, they calculated a range of 5 to 65 nmol/L, while Kjølbye et al. suggested a range of 5 to 15 nmol/L [40,41]. All of these are considerably below that of the AGNP.

Although, in general, the closest correlation to the published therapeutic reference ranges seemed to be achieved by comparison to 10–90 percentiles, a nearly perfect match was obtained for amitriptyline + nortriptyline (total), aripiprazole, dehydroaripiprazole, aripiprazole + dehydroaripiprazole (total), and dosulipine when the same calculation was performed using 25–75 percentiles (Table 3). For amitriptyline + nortriptyline (total), the main references for the range in the AGNP guideline seems to be that of Perry et al. and that of Vandel et al. [42,43]. For aripiprazole, dehydroaripiprazole, and aripiprazole + dehydroaripiprazole (total), several of the listed references herein are population-based studies where the therapeutic reference range is calculated as 25–75 percentiles [44,45,46]. In contrast, the citations given for the therapeutic reference range of dosulipine are all older studies (>25 years) and not population based [3].

### 3.4. Investigation of Age and Sex Differences

Sex difference in the calculated therapeutic analytical ranges was investigated by using the built-in feature of the RefIT software. The combined and sex-specific therapeutic analytical ranges are shown in Table 4. In general, ranges calculated on the basis of data from women alone were higher than those from men. The only exception was for the antidepressant escitalopram.

To investigate the effect of age, therapeutic analytical ranges were calculated on the basis of two intervals: 20–64 years and 65–100 years (Table 5). Only minor differences could be observed for most of the therapeutic drugs, and no consistent trend was identified between the age groups.

## 4. Discussion

The purpose of the presented work was to compare routine laboratory data sets from multiple laboratories and evaluate their correspondence to published therapeutic reference ranges. As the data for calculating a therapeutic reference range rely on the study design, population type and size, dosage, and precision of the analytical method, conducting a direct comparison is challenging [7]. Additionally, in contrast to endogenous substances, there is no consensus method for the inclusion of patients or for calculating therapeutic reference ranges for drugs [3,7,47]. While a 10–90% range is commonly applied for population-based studies, many report a 25–75 percentile [8,41]. The AGNP consensus guideline does not clearly distinguish between them, and in addition, many of the listed ranges seem to have been concatenated from several studies (both population-based and controlled setups) [11]. In our experience, the use of a lower fraction can lead to too-narrow ranges. As measurements from the same patient may vary considerably even when all samples are taken at the minimum drug concentration at steady state, this can cause reason for untimely adjustment of dosage. Because of this, we opt for the use of 10–90% ranges, while for comparison, we have included values for the 25–75 percentiles (Table 2 and Table 3).

Although the developed TDM model allows the objective selection of data when calculating therapeutic analytical ranges, it does not provide the complete removal of all unwanted subpopulations from a data set. As the last sample is always included, the data set is presumed to contain results from patients who have had the treatment terminated for various reasons. By allowing the inclusion of more than one sample per patient, the model tries to correct for the effect of these, by enriching the data set for patients in long-term treatment. The model also increases the total number of included samples, thus obtaining better statistic support from a small data set compared with using only a single result per patient (Table 1).

The calculation of therapeutic analytical ranges from different data sets facilitated a comparison of the analytical methods used by the participating laboratories. In general, there was good concordance between them, except for three of the analytes. These were citalopram/escitalopram, mirtazapine, and quetiapine. The analysis of mirtazapine was discontinued in 2018 at the EHL owing to interference issues that could not be resolved on the standard equipment at that time. This discrepancy was therefore anticipated (Table 2). The deviations observed for citalopram/escitalopram may be attributed to variations from using a small data set. The low number of patients here may also suggest that the results have been requested as part of compliance or intoxication test rather than routine TDM monitoring. However, the comparison of the larger data sets shows the strength of monitoring the trend in patient data and using these for interlaboratory comparison. This because in contrast to regular internal quality controls and proficiency testing, it monitors over time the trend in real patient samples and matrices and provides a test of the whole environment of the analysis, from blood sampling to analytical instrument and calibration, through middleware, and to the final report in LIMS. An automated approach as presented here may thus provide a fast comparison between laboratories without exchanging samples.

Although the retrospective laboratory data from a TDM analysis come without any prior knowledge of dosage, comedication, reason why a test has been ordered, anamnesis of the patient, or therapeutic outcome, the data have the benefit of representing a naturalistic population rather than a specifically selected study cohort. Therefore, this range reflects all the factors which may influence the result of the measurement. While the upper limit of the therapeutic analytical range is influenced primarily by dosage and interpersonal variations, the lower 10 percentile may be influenced more by additional factors. These factors include the limit of the quantification of the analysis, the number of patients not in compliance, the prolonged fasting period prior to sampling, and the placebo effect where a clinical response is observed, at what would normally be at an insufficient dosage. Because of this, the deviation of the calculated lower limit of the therapeutic analytical range is expected to be larger than that for the upper limit, and correlates less to that of the therapeutic reference range. This may explain why the lower limits of the therapeutic analytical ranges were in general found to be below those of the published therapeutic reference ranges (Table 2). Although this held true for most of the drugs in this study, exceptions were observed for flupentixol, haloperidol, paroxetine, perphenazine, and total venlafaxine + o-desmethyl-venlafaxine. Here, both the calculated 10% and 90% limit of the percentiles were higher than the ranges published in the AGNP consensus guideline. In contrast, sertraline, ziprasidone, and zuclopenthixol each had an upper 90% range that was markedly lower than these. It is interesting to note that the drugs flupentixol, haloperidol, perphenazine, and zuclopenthixol all are quantified at the lower (<100) nmol/L level in plasma. As quantification at this scale even with modern equipment can be troublesome. This may explain some of the discrepancy as many of the therapeutic reference ranges dates back to studies performed in the 1980s, and are based on a single analytical method. However, the difference may also be due to the naturalistic setting of the data sets in this study, compared with a controlled population. Both perphenazine and zuclopenthixol have been found to be strongly influenced by the metabolic status of the CYP2D6 enzyme [48,49,50,51,52,53,54,55,56]. For these drugs, patients prescribed a standard dosage and who are either poor metabolisers or subject to drug interactions are expected to harbour a higher serum concentration. The higher upper fraction of the therapeutic analytical ranges in this study may thus reflect such patients in the data sets that are treated with a standard dosage.

Our data on paroxetine, which are supported by the findings of Reis et al., showed a higher upper therapeutic analytical range than the reference provided in the AGNP [3,8]. This is interesting as a negative effect on the treatment at higher dosage has been shown for this drug [57]. The higher upper limit of the calculated therapeutic analytical range may suggest that a relatively large number of the patients receiving paroxetine may be above the limit where there is negative response to the treatment [57,58]. It is therefore important that this difference is resolved.

Using the built-in features of the RefIT software, we investigated the effect of sex and age on the calculated therapeutic analytical ranges. Consistent with previous reports, women in general had higher blood concentrations than did men. This was true for all the investigated drugs, except for the antidepressant escitalopram (Table 4). Escitalopram is the stereo enantiomer of citalopram, and both are metabolised by the same CYP p450 enzymes, namely CYP2C19, CYP3A4, and to a lesser extent CYP2D6. Although previous studies have indicated that women have higher concentrations of some metabolites of escitalopram and higher activity of CYP3A4 than men do, studies conducted for the labelling of the drug did not show any sex difference [59,60]. However, as there are more results for women in the data set than for men, the difference could also be due to statistical uncertainty when applying the model.

Age is a second well-known factor influencing pharmacokinetics, with the primary reason being an acquired lower blood flow through the liver with ageing [61,62,63,64]. Comparing the calculated therapeutic analytical ranges for two age groups, 20–64 and 65–100 years, did not show any consistent differences between the results for the analysed drugs (Table 5). A possible explanation for this might be that the data used for the calculation were not corrected for dosage, and most patients in the data set were titrated to diminish the effect of age.

Although the calculated therapeutic analytical ranges do not provide any information on the effect of the treatment, a certain correlation with the therapeutic reference range is anticipated. The current study therefore shows that literature data cannot be directly transferred into a routine laboratory setting without validation. Furthermore, it highlights the importance of periodically re-evaluating laboratory therapeutic reference ranges as treatment paradigms may change over time [12]. Developing a consensus method for the calculation, as well as software capable of automatically handling large data sets, can help establish more reliable and thoroughly validated therapeutic reference ranges.

## 5. Conclusions

In this study, we designed a model for selecting and cleaning routine laboratory data. We then applied the algorithm for calculating therapeutic analytical ranges from the TDM data sets of antidepressants and antipsychotics from three laboratories. The analysis supported the therapeutic reference ranges published in the AGNP consensus guideline for the drugs amitriptyline + nortriptyline (total), nortriptyline, aripiprazole, aripiprazole + dehydroaripiprazole, citalopram, escitalopram, clomipramine + desclomipramine (total), clozapine, duloxetine, fluoxetine + norfluoxetine (total), imipramine + desipramine (total), mirtazapine, olanzapine, paliperidone, quetiapine, risperidone + paliperidone (total), and sertindole. In contrast, major deviations were observed when comparing laboratory data with the therapeutic reference ranges for the drugs desipramine, flupentixol, haloperidol, paroxetine, perphenazine, venlafaxine + o-desmethyl-venlafaxine (total), sertraline, ziprasidone, and zuclopenthixol. Support for our findings was found in other studies than those cited by the AGNP, and we therefore opt for a re-examination of these. Our model and our algorithm provide the first step of automation and present a tool to support the clinical validation of therapeutic reference ranges based on big data [10].

## Figures and Tables

**Figure 1 pharmaceutics-15-00673-f001:**
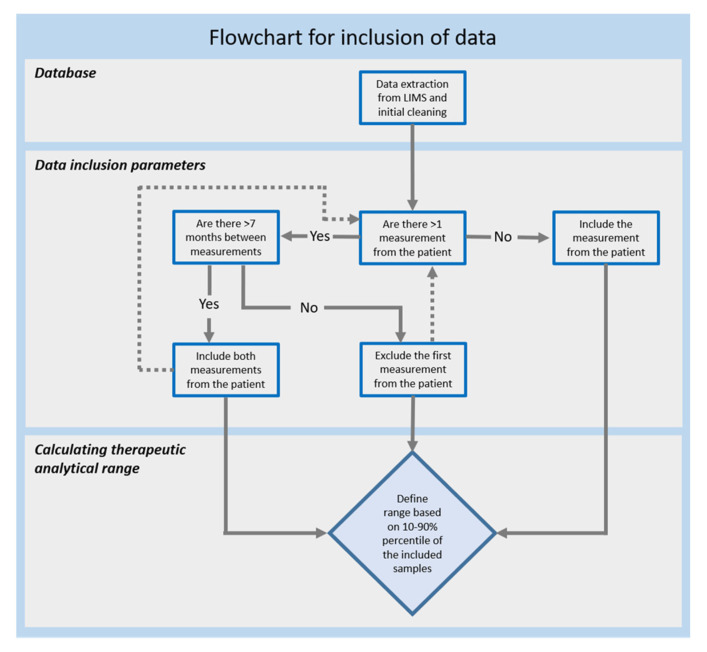
Flowchart of the TDM model used for qualification of data for calculating therapeutic analytical ranges.

**Figure 2 pharmaceutics-15-00673-f002:**
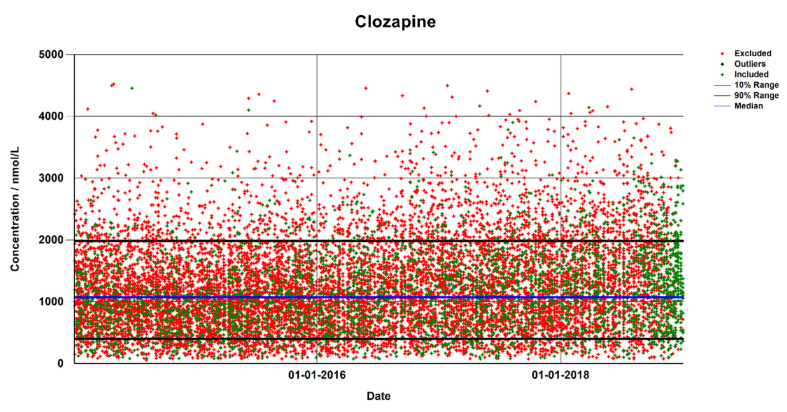
Plot of data points from the data set of clozapine from Aarhus University Hospital. Red dots are the sample results excluded and green are the samples included by the TDM model. The median is shown as a blue line, while the 10 and 90 percentiles are shown in black. The percentiles and median were calculated on the basis of the included samples only (included *n* = 3224, excluded *n* = 10,871).

**Figure 3 pharmaceutics-15-00673-f003:**
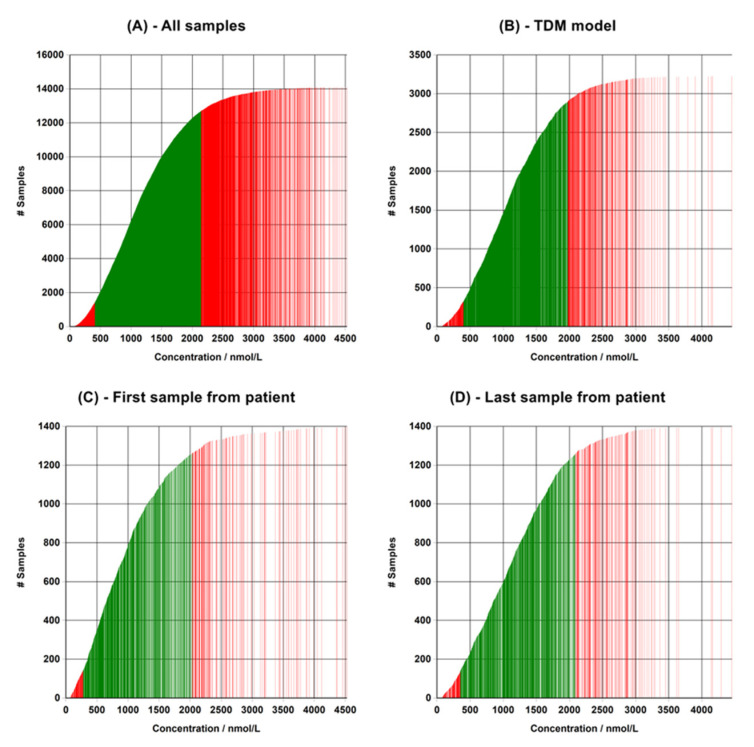
Evaluation of the described TDM model for selecting results from retrospective laboratory data sets. Graphs shows percentile plots for clozapine as examples, where each line represents a sample result. (**A**) All samples in the data set, (**B**) samples selected using the TDM model, (**C**) only the first result from a patient is used, and (**D**) only the last result from a patient is used. The 10–90% percentile is marked in green. The graphs were generated and exported by using the RefIT software.

**Table 1 pharmaceutics-15-00673-t001:** Therapeutic analytical ranges for clozapine, perphenazine, and imipramine calculated based on the data set from Aarhus University Hospital.—(A) all samples in the data set, (B) samples selected by using the TDM model, (C) only the first result from a patient, and (D) only the last result from a patient.

	Clozapine Patients = 1398		Perphenazine Patients = 405		Imipramine Patients = 127	
	Incl. Samples	Range nmol/L	Incl. Samples	Range nmol/L	Incl. Samples	Range nmol/L
(A) All samples	14,095	409–2143	1224	1.5–14.3	281	43–846
(B) TDM model	3224	400–1980	621	1.8–13.8	146	27–437
(C) First sample	1398	276–2034	405	1.0–14.4	127	25–331
(D) Last sample	1398	354–2091	405	1.5–13.6	127	26–390

**Table 2 pharmaceutics-15-00673-t002:** Comparison between therapeutic reference ranges listed in the AGNP consensus guideline and calculated 10–90 percentiles from the three laboratories. The data were selected on the basis of the TDM model, and ranges were calculated by using the RefIT software. The number of patients is shown under *p*. Abbreviations: ND—no data, AUH—Aarhus University Hospital, EHL—the Danish Epilepsy Centre, and RH—Rigshospitalet. Total number of samples, number of included samples used for the calculation of percentiles, and additional statistics are supplied as Appendix A.

Therapeutic Drug		AUH		EHL		RH	
	AGNP	*p*	Range nmol/L	*p*	Range nmol/L	*p*	Range nmol/L
Amitriptyline	ND	1033	35–464	612	83–479	1492	20–441
Nortriptyline	266–646	4436	105–623	2016	104–630	4671	69–661
Amitriptyline + metabolite	288–720	1308	79–809	570	201–917	1025	129–906
Aripiprazole	223–781	1695	116–1148	1449	154–1064	ND	-
Dehydroaripiprazole	ND	1662	42–347	1366	83–406	ND	-
Aripiprazole + metabolite	335–1115	1889	217–1378	1365	283–1465	ND	-
Citalopram	154–339	1981	69–379	632	80–499	ND	-
Escitalopram	46–246	795	33–197	99	31–283	ND	-
Dosulipine	153–339	ND	-	ND	-	107	20–505
Northiaden	ND	ND	-	ND	-	105	20–356
Dosulipine + metabolite	ND	ND	-	ND	-	92	119–888
Clomipramine	ND	1192	96–610	400	122–628	448	71–652
Desmethylclomipramine	ND	1185	82–777	394	136–895	445	45–950
Clomipramine + metabolite	731–1494	1156	285–1252	394	326–1445	409	272–1497
Clozapine	1071–1836	1398	400–1980	1355	349–2389	2028	130–2180
Duloxetine	100–403	1467	47–428	179	47–421	ND	-
Fluoxetine	ND	468	118–1183	ND	-	ND	-
Norfluoxetine	ND	472	255–1169	ND	-	ND	-
Fluoxetine + metabolite	388–1695	326	356–1486	ND	-	ND	-
Flupentixol	1.2–11.5	ND	-	136	2–23	ND	-
Haloperidol	2.7–26.6	ND	-	445	5–48	ND	-
Imipramine	ND	127	27–437	95	71–530	178	20–588
Desimipramine	375–1125	126	20–409	88	73–510	179	20–528
Imipramine + metabolite	641–1098	124	68–812	79	175–1032	136	141–1137
Mirtazapine	113–302	1000	37–256	253	75–398	ND	-
Olanzapine	64–256	1854	38–247	2663	30–273	ND	-
Paroxetine	61–198	ND	-	268	61–537	ND	-
Perphenazine	1.5–6	405	1.8–13.8	360	2–18	ND	-
Quetiapine	261–1305	2653	33–851	1649	48–1268	ND	-
Risperidone	ND	1564	3–58	880	7–82	ND	-
Paliperidone	47–141	2349	11–109	959	17–121	ND	-
Risperidone + metabolite	41–146	2098	16–139	851	35–177	ND	-
Sertindole	114–227	ND	-	114	48–236	ND	-
Sertraline	33–491	3438	33–233	934	36–289	ND	-
Venlafaxine	ND	4154	57–966	2938	96–1073	ND	-
O-Desmethyl-venlafaxine	ND	4181	309–1774	3107	312–1694	ND	-
Venlafaxine + metabolite.	361–1520	4134	489–2522	2926	615–2588	ND	-
Ziprasidone	128–510	246	38–347	ND	-	ND	-
Zuclopenthixole	10–125	885	6.5–42.8	1062	7–62.2	ND	-

**Table 3 pharmaceutics-15-00673-t003:** Comparison of calculated 25–75 percentiles between the participating laboratories and to the therapeutic reference ranges published in the AGNP consensus guideline. For abbreviations, see Table 2. Additional statistics is in Appendix A.

		AUH		EHL		RH	
Therapeutic Drug	AGNP	*p*	Range nmol/L	*p*	Range nmol/L	*p*	Range nmol/L
Amitriptyline	ND	1033	77–310	612	140–332	1492	20–281
Nortriptyline	266–646	4436	221–504	2016	195–476	4671	195–502
Amitriptyline + metabolite	288–720	1308	180–596	570	304–684	1025	245–676
Aripiprazole	223–781	1695	266–788	1449	279–726	ND	-
Dehydroaripiprazole	ND	1662	93–255	1366	123–294	ND	-
Aripiprazole + metabolite	335–1115	1889	392–1035	1365	448–1040	ND	-
Citaloprame	154–339	1981	113–271	642	133–336	ND	-
Escitaloprame	46–246	795	50–133	99	54–134	ND	-
Dosulipine	153–339	ND	-	ND	-	107	100–347
Northiaden	ND	ND	-	ND	-	105	44–206
Dosulipine + metabolite	ND	ND	-	ND	-	92	203–578
Clomipramine	ND	1192	188–444	400	191–439	448	167–444
Desmethylclomipramine	ND	1185	188–575	394	231–603	445	189–637
Clomipramine + metabolite	731–1494	1156	477–1011	394	477–990	409	474–1127
Clozapine	1071–1836	1398	677–1528	1355	656–1750	2028	478–1502
Duloxetine	100–403	1467	86–278	179	83–259	ND	-
Fluoxetine	ND	468	249–771	115	363–989	ND	-
Norfluoxetine	ND	472	421–845	112	401–764	ND	-
Fluoxetine + metabolite	388–1695	326	584–1100	110	838–1895	ND	-
Flupentixole	1.2–11.5	ND	-	136	3–13	ND	-
Haloperidole	2.7–26.6	ND	-	445	9–31	ND	-
Imipramine	ND	127	71–253	95	100–324	178	68–394
Desimipramine	375–1125	126	41–249	88	97–347	179	55–327
Imipramin + metabolite	641–1098	124	128–535	79	262–632	136	254–857
Mirtazapine	113–302	1000	64–178	253	106–242	ND	-
Olanzapine	64–256	1854	65–167	2663	58–178	ND	-
Paroxetine	61–198	ND	-	268	114–321	ND	-
Perphenazine	1.5–6	405	3.2–8.4	360	4–11	ND	-
Quetiapine	261–1305	2653	79–490	1649	125–709	ND	-
Risperidone	ND	1564	6–28	880	10–38	ND	-
Paliperidone	47–141	2349	23–73	959	30–82	ND	-
Risperidone + metabolite	41–146	2098	31–95	851	53–123	ND	-
Sertindole	114–227	ND	-	114	78–160	ND	-
Sertraline	33–491	3438	58–153	934	62–179	ND	-
Venlafaxine	ND	4154	133–575	2938	165–645	ND	-
O-Desmethyl-venlafaxine	ND	4181	545–1328	3107	530–1267	ND	-
Venlafaxine + metabolite.	361–1520	4134	815–1901	2926	883–1921	ND	-
Ziprasidone	128–510	246	77–237	ND	-	ND	-
Zuclopenthixole	10–125	885	10.5–29.5	1062	13–41	ND	-

**Table 4 pharmaceutics-15-00673-t004:** Comparison between calculated 10–90 percentile therapeutic analytical ranges for men and women performed on the data set from Aarhus University Hospital. Here, *p* indicates the number of patients that the calculations are based on. For additional statistics, see Appendix A.

	Men		Women	
Therapeutic Drug	*p*	Range nmol/L	*p*	Range nmol/L
Amitriptyline	366	34–423	666	36–473
Nortriptyline	1618	98–613	2817	111–629
Amitriptyline + metabolite	446	76–753	861	81–835
Aripiprazole	797	91–1122	897	138–1197
Dehydroaripiprazole	784	35–334	877	49–356
Aripiprazole + metabolite	883	201–1354	1005	237–1400
Citalopram	671	66–331	1309	70–392
Escitalopram	269	34–213	526	32–186
Clomipramine	457	81–606	734	110–613
Desmethylclomipramine	454	69–714	730	90–823
Clomipramine + metabolite	441	241–1183	714	307–1276
Clozapine	818	380–1930	578	428–2030
Duloxetine	446	38–373	1020	50–438
Fluoxetine	124	83–836	344	138–1255
Norfluoxetine	125	206–1065	347	291–1223
Fluoxetine + metabolite	94	323–1215	232	380–1633
Mirtazapine	433	36–243	567	39–262
Olanzapine	1077	38–233	776	38–272
Perphenazine	181	1.7–12.3	224	1.8–15.3
Quetiapine	1098	37–857	1553	31–847
Risperidone	867	3–57	696	4–59
Paliperidone	1355	11–103	993	11–118
Risperidone + metabolite	1202	16–127	895	17–157
Sertraline	1108	31–223	2330	34–236
Venlafaxine	1373	51–840	2780	61–1043
O-Desmethyl-venlafaxine	1381	286–1699	2799	316–1798
Venlafaxine + metabolite.	1369	462–2414	2764	517–2582
Ziprasidone	88	34–322	157	40–376
Zuclopenthixole	489	7.4–41.2	396	6.1–44.6

**Table 5 pharmaceutics-15-00673-t005:** Examination of the influence of age on the calculated 10–90 percentile therapeutic analytical range. Ranges of 20–64 and 65–100 years were calculated with the RefIT software by using the data set from Aarhus University Hospital. Here, *p* indicates the number of patients included in the data set of each compound. ND—no data; <100 samples. For additional statistics, see Appendix A.

	Age 20–64		Age 65–100	
Therapeutic Drug	*p*	Range nmol/L	*p*	Range nmol/L
Amitriptyline	772	34–454	250	39–471
Nortriptyline	3510	107–625	947	97–614
Amitriptyline + metabolite	1214	77–800	328	87–836
Aripiprazole	1424	135–1154	108	1–1196
Dehydroaripiprazole	1405	46–354	96	12–380
Aripiprazole + metabolite	1590	211–1381	102	221–1409
Citalopram	1408	65–363	554	78–409
Escitalopram	607	32–188	188	36–219
Clomipramine	996	99–623	196	83–560
Desmethyl-clomipramine	987	85–775	197	70–804
Clomipramine + metabolite	964	284–1252	192	314–1234
Clozapine	1272	411–1999	137	333–1793
Duloxetine	1154	43–402	290	63–468
Fluoxetine	269	111–1210	ND	-
Norfluoxetine	270	246–1119	ND	-
Fluoxetine + metabolite	215	327–1650	ND	-
Mirtazapine	615	34–235	376	42–280
Olanzapine	1497	39–256	323	35–204
Perphenazine	335	1.6–14	69	2.1–12.7
Quetiapine	2152	33–877	352	31–699
Risperidone	1238	3–62	270	3.3–42
Paliperidone	1883	12–111	353	11–115
Risperidone + Paliperidone	1647	18–141	341	18–142
Sertraline	2650	32–232	402	32–231
Venlafaxine	3445	54–934	701	76–1084
O-Desmethyl-venlafaxine	3470	295–1699	703	364–1973
Venlafaxine + metabolite	3430	468–2460	697	683–2774
Ziprasidone	230	40–361	ND	-
Zuclopenthixol	729	6.9–44.1	164	5.5–35.5

## Data Availability

Data are contained within the article or Appendix A.

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
