# Peer review of "Automated Interlaboratory Comparison of Therapeutic Drug Monitoring Data and Its Use for Evaluation of Published Therapeutic Reference Ranges"

_pharmaceutics, 2023, doi:10.3390/pharmaceutics15020673_

Round 1

Reviewer 1 Report

In the manuscript entitled “Automated inter-laboratory comparison of therapeutic drug monitoring data and its use for evaluation of published therapeutic reference ranges” Borggaard Larsen J. et colleagues presented a new method, in-house software based, for comparing published therapeutic reference ranges for antidepressant and antipsychotic drugs with real life patient’s data collected in different hospitals.

The publication by Hiemke et colleagues (1) was chosen as the gold standard, which is also adopted by most European laboratories and diagnostic kit manufacturers. The authors themselves add that in some cases they had to refer to other publications for a critical review of the published values.

In my opinion, there are two weak points in the manuscript, emphasised by the authors themselves: the fact that the methods for collecting and analysing these ranges are often not described in the reference publications or that they are given as references by other authors within (1); furthermore, the adoption of the term 'therapeutic ranges' may be of some concern.

Indeed, the problem is not new and is still an open discussion.

I believe that, in order to be able to define reference ranges as being therapeutic ranges in addition to what is reported in the CLSI document EP28-A3c (2), it would be necessary to take into account, above all, the drug efficacy measured as patient's outcome. In fact, ref (2) suggests in the specifics of determining therapeutic ranges of drugs that: " This guideline does not address the determination of therapeutic drug levels. This requires a different study. The population required for these studies necessarily has to be under the influence of the pharmacologic agent and at a clinically effective level. This problem is complex and involves a number of additional issues such as dosage, dosing, time of specimen procurement in relation to time of administration of the drug, the route of drug administration, clinical effectiveness, toxicity, and other issues."

Unfortunately, this is a very difficult job from which ref (1) also suffers althoug it represent a "gold standard" for most laboratories worldwide. With the adoption of the software developed by the authors, an attempt is made to overcome this problem by introducing the variables "time between sample dates" and the adoption of expedients such as various evaluation methods (TDM model, last result, first result). The results thus obtained have strong statistical value with lower clinical value. However, the approach is extremely interesting and should definitely be developed. I believe it is necessary in the study of therapeutic ranges to also introduce the study of interactions with other drugs (a feature of the software still under development). Indeed, clinical practice has shown over the years, for a considerable number of drugs, how these ranges are strongly influenced by comedications.

I would therefore ask the authors to further stress these issues, which I repeat, have not been left out or hidden, by also introducing a reference to what is reported in ref. (2).

The manuscript is certainly interesting and paves the way for a new way of addressing this issue and certainly deserves to be published in Pharmaceutics.

1) Hiemke, C.; Bergemann, N.; Clement, H.W.; Conca, A.; Deckert, J.; Domschke, K.; Eckermann, G.; Egberts, K.; Gerlach, M.; 562 Greiner, C.; et al. Consensus Guidelines for Therapeutic Drug Monitoring in Neuropsychopharmacology: Update 2017. Phar-563 macopsychiatry 2018, 51, 9-62, doi:10.1055/s-0043-116492.

2) Clinical Laboratory Standards Institute (CLSI). Defining, establishing, and verifying reference intervals in the clinical laboratory; approved guideline, 3rd ed. CLSI document EP28-A3c. Wayne, PA, 2010.

The title is too general since data regarding only antidepressants and antipsychotics are used. I therefore urge the Authors to reformulate the title by highlighting this fact.

Author Response

The author’s appreciates the thoroughness of the reviewer, and insight into the difficulties and complexity of defining and validating therapeutic reference ranges. A fact we are well aware of and have made a point of describing in the manuscript.

We are also happy to read that the reviewer share our vision for the presented work and the algorithm we have developed. It is correct that we are focusing on psychopharmacas (antidepressants and antipsychotics). However, the methodology is applicable for any use of TDM, including antiepileptica, immunosuppressiva, anticoagulants, and antibiotics. Indeed, as the software also output ranges calculated as +/-2SD of a dataset, the software can be used on endogenous substances as well. Although our approach does not solve all the hurdles, we believe it is a step forward for therapeutic drug monitoring, allowing big-data to be used in the evaluation of therapeutic reference ranges. 

Our choice of title is made to reflect this, as it promotes the methodology rather than the drugs it is applied upon, and we sincerely believe the title accommodates precisely our intentions with the manuscript. Also, although we would like to mention in the title that we use the method on antidepressiva and antipsychotic, we cannot see how to include this without the title losing one of the existing points or getting excessively long. As we want to target a broad audience, we hope instead that readers only seeking information regarding therapeutic reference ranges of psychopharmacas, will read though the abstract and hereby become interested in the work.

For these reasons we plea that the title is maintained as it is. However, we appreciate the comment from the reviewer, and have tried to accommodate her/his opinion by making a change to the introduction in the abstract (line 18-19), focusing it more on the general use of TDM.

As indicated in the software by the ‘co-medication’ feature (in development), we have made similar thoughts as the reviewer, about the effects of pharmacogenetics and drug-drug interactions, and including this in a future release of the software. However, for now we find it outside the scope of the presented work.

We have included the reference specified by the reviewer, and added it to line 76, 504 and 506 (reference [7]).

We hope these changes are acceptable.

Sincerely

The Authors

Reviewer 2 Report

An interesting approach to validate the therapeutic range for neuropsychopharmacological drugs. There are some limitations which are mentioned by the authors. 

Therapeutic ranges in TDM are problematic especially in this field and this study adds another piece for improvement.

Author Response

We are very happy that the reviewer share our vision for the developed methodology. As mentioned in the review, we are well aware of the limitations, and acknowledge this in the manuscript. We thank the reviewer for the positive feedback.

Sincerely

The Authors

Reviewer 3 Report

Please replace "gender" with "sex" throughout the text (including tables/figures if applicable), if the authors referred to biological sex.

Author Response

We have exchanged the use of the word ‘gender’ with ‘sex’ throughout the manuscript, and are thankful for the positive review.

Sincerely

The Authors

Reviewer 4 Report

The authors present an approach that compares laboratory datasets within the context of TDM and evaluate their correspondence to generally clinically applied or aimed therapeutic reference ranges for 31 antidepressant and antipsychotic drugs. The work is interesting, well-structured and generally well presented. I have a minor comment that the work focuses mostly in CNS drugs so this may should be reflected with the title becasue TDM can take place also for cardiovascular drugs, warfarin etc. etc. Moreover, I would like to see more about CYP polymorphisms and TDM and maybe some drug-drug interactions (?) that could modulate the concentrations of drugs near the top or bottom ranges. But the authors provide a small sentence. It would be interesting if, according to authors, should be considered within the automated approach. Apart of those minor comments the manuscript can be accepted as it is. Congratulations for the very good work. 

Author Response

We are very grateful of the positive review. The reviewer has two comments; One regarding the inclusion of pharmacogenetic variability and drug-drug interactions in the datasets, and the second on the title of the manuscript.

The use of pharmacogenetic testing in relation to dosing guidelines is an ongoing discussion. We agree with the reviewer that this in theory can influence the calculated ranges, depending on the pharmacokinetic of the drug. However, this effect may be diminished as any adverse reaction or low/high TDM levels in the model is expected to trigger a correction of dosage and a new measurement. We acknowledge that it would be interesting to incorporate the genotype of patients into the model/software, but find that it, for now, is beyond the scope of the manuscript. Instead we have tried to include this view, by adding genetic variations to the list of subpopulations, line 148, “Patients receiving standard dosage, but who are pharmacogenetic poor or ultrafast-metabolisers of the drug”.

Regarding the title and the request to include which drugs the study focus on. We acknowledge the point of the reviewer, but find it difficult to accommodate. As the methodology used in the manuscript is applicable for any use of TDM, including antiepileptica, immunosuppressiva, anticoagulants, and antibiotics, we would like to promote this in the title. By focusing the use on psychopharmacas only, this point may be lost. We do appreciate the comment of the reviewer, and have tried to include it by changing the introduction of the abstract (line 18-19), hereby drawing the attention away from psychiatric drugs and towards the general use of the developed algorithm. We hope that these changes are acceptable.

Sincerely

The Authors